# Relaxed Structure Tensor Representation for Robust Oriented Object Detection

## Abstract

Oriented object detection predicts oriented bounding boxes. Precisely predicting their orientation remains challenging due to angular periodicity, which introduces boundary discontinuity issues and symmetry ambiguities. In this paper, we introduce Relaxed Structure Tensor Bounding Boxes (RST-BB), a representation inspired by classical image structure tensors encoding object orientation in addition to height and width. RST-BB provides a simple yet efficient angle-coder approach that is robust to angular issues, effectively addresses square objects, and requires no additional hyperparameters. Extensive evaluations across five datasets demonstrate that RST-BB achieves state-of-the-art results with high angular prediction precision, establishing relaxed structure tensors as a robust and modular alternative for encoding orientation in oriented object detection. We make our code publicly available for seamless integration into existing detectors.

## 1 Introduction

Object detection is one of the classical problems in computer vision and traditionally localizes objects with horizontal bounding boxes (HBB) (Zhao et al., 2019). Nevertheless, certain areas such as aerial imagery (Xia et al., 2018; Sun et al., 2022; Yang & Yan, 2022; Yang et al., 2023a; Yu et al., 2024b) and scene text detection (Liao et al., 2018; Liu et al., 2018; Ma et al., 2018; Wang et al., 2020) require information on object orientation for more accurate identification (Zhou et al., 2022). Hence, oriented object detection extends object detection by predicting oriented bounding boxes (OBBs) that better align with object boundaries.

Despite progress in recent years, angular periodicity still poses challenges when predicting object orientation. In the first place, small variations between the prediction and ground truth at the angular boundary will cause a sharp loss increase. This phenomenon, known as the boundary problem (Xiao et al., 2024; Yang & Yan, 2020; Xu et al., 2024), can penalize the network while comparing mathematically similar values. In addition, objects with axial symmetry can cause the network to penalize equivalent angle predictions. For example, $\frac{\pi}{2}$ and $-\frac{\pi}{2}$ radians are equivalent orientations for rectangular objects, but their loss would be significantly high (Xiao et al., 2024; Yu & Da, 2023).

Previous works have addressed these discontinuity problems in different ways. Some approaches propose angle-coder solutions (Yang & Yan, 2020; Yang et al., 2021a; Yu & Da, 2023), which transform orientation into representations that circumvent angular periodicity issues. However, their performance can be greatly affected by hyper parameters (Yu & Da, 2023; Xiao et al., 2024). Other methods mitigate these issues by representing OBBs as 2D Gaussian distributions (Yang et al., 2021c;d; 2023b), and compare the distance between these representations using distribution-based computations. They elegantly provide a continuous orientation representation, though they still introduce hyperparameters, which can make their loss unstable and difficult to tune across datasets (Yu & Da, 2023; Yang et al., 2023b), and struggle to handle square-like objects. Furthermore, the improvements reported focus on mean average precision (mAP), without specifically evaluating the precision of orientation predictions.

Traditional image analysis methods, particularly for edge and corner detection (Harris et al., 1988; Förstner & Gülch, 1987; Lindeberg, 1998; Brox & Weickert, 2006; Scharr, 2004), have investigated the representation of orientation, shape, and symmetry through the concept of structure tensor (Bigun et al., 1991). A structure

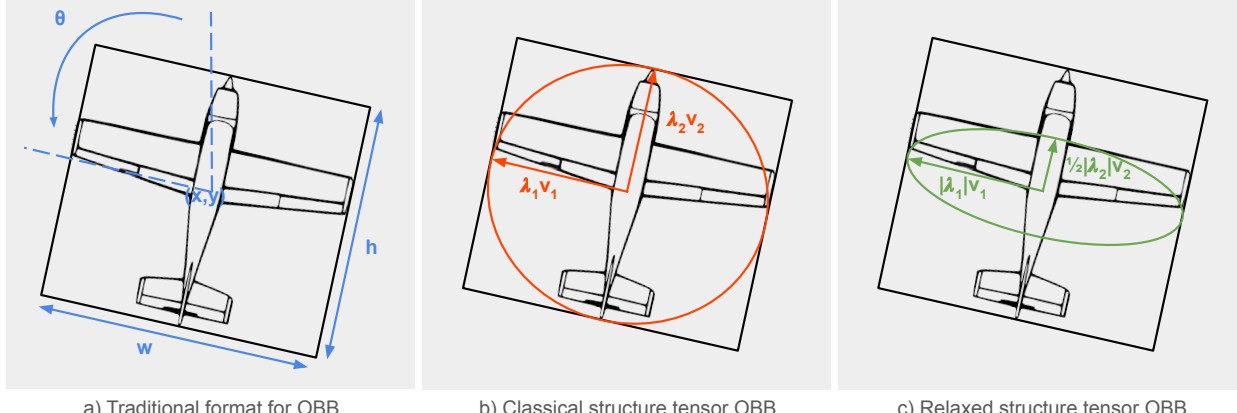

a) Traditional format for OBB     b) Classical structure tensor OBB     c) Relaxed structure tensor OBB

Figure 1: **Relaxed Structure Tensor Bounding Boxes (RST-BB).** Comparison of traditional OBBs (blue), classical structure tensors (red), and relaxed structure tensors (green) for square-like objects. a) The traditional $(x, y, w, h, \theta)$ OBB format suffers from boundary issues and ambiguity, with four possible solutions for squares. b) Modeling OBBs with classical structure tensors, similar to Gaussian-based approaches, avoids this but yields no clear principal direction for square-like objects. c) To address this, we propose relaxed structure tensors, leveraging the fact that $w \geq h$ and ensuring a unique solution with a strong principal orientation.

tensor is a $2 \times 2$ symmetric matrix that encodes information of the orientation and anisotropy of a local structure. Structure tensors offer two key advantages: 1) they are inherently continuous over the angular space, and thus are robust to boundary discontinuities, and 2) their flexibility to model different symmetries through their eigenvalues. Moreover, calculating structure tensors is straightforward and computationally efficient, which can be done in an angle-coder fashion. Thus, as illustrated in Figure 1, we propose to represent the orientations of objects as symmetric $2 \times 2$ matrices, which adapt classical structure tensors to encode orientation and width and height of objects. Since we will not require the positive semi-definitiveness property of classical structure tensors, we refer to the resulting bounding boxes as Relaxed Structure Tensor Bounding Boxes (RST-BB), which combine the simplicity and modularity of angle-coder methods with the performance of Gaussian-based approaches. Our representation is robust to boundary issues, symmetric ambiguities, and square objects, requiring no additional hyperparameters.

We perform extensive experiments and compare the performance of our approach with previous work. We show that the Relaxed structure tensor representation can considerably reduce the orientation prediction error while achieving mAP state-of-the-art (SOTA) performance across five different datasets. Our contributions can be summarized as follows.

- To the best of our knowledge, this is the first work to propose a representation of orientation in OBBs as relaxed structure tensors. A simple and modular implementation is made publicly available, enabling its integration into any detector.

- Through extensive experimentation, we show that our approach is able to decrease the orientation prediction error, while maintaining SOTA performance. In addition, we effectively handle square objects, unlike previous methods.

- We combine angle-coder and Gaussian-based strengths into a robust and modular solution that elegantly addresses boundary problems. Without requiring additional parameters, our solution can be easily integrated into existing detectors.

## 2 Related works

**Object Detection** is a classical computer vision task that aims to identify instances of visual objects in images and accurately estimate their location, providing an important piece of information: *What objects are where?* (Zou et al., 2023; Zhao et al., 2019). Like in other fields, deep neural networks (DNN) have brought about large improvements in object detection, and DNN-based approaches have become the state of the art over classical approaches (Xiao et al., 2020). Modern object detectors can be grouped into one-stage detectors and two-stage detectors. One-stage detectors localize potential objects over a grid of locations and determine their category in one single step (Redmon et al., 2016; Lin et al., 2017; Howard et al., 2017). Instead, two-stage detectors first extract a number of class-agnostic bounding box proposals using a region proposal network. Then, a sub-network classifies these region proposals among all possible object categories (Girshick et al., 2014; Ren et al., 2015; He et al., 2017). While one-stage detectors provide a more time-efficient solution by merging regression and classification into one single step, two-stage detectors generally achieve higher performance. Regardless of their type, early DNN-based detectors considered dense sampling of locations, known as anchors, to predict bounding boxes in the image. However, later works have proposed anchor-free approaches in order to match the performance of two-stage detectors without the computational cost of anchor-based regression (Law & Deng, 2018; Duan et al., 2019; Tian et al., 2019; Yang et al., 2019b). More recently, Carion et al. (2020) introduced DETR, a transformer-based method that detects objects using a fixed set of learnable queries. The concept of query-based detection has had considerable impact in the field, inspiring a progression of recent works (Chen et al., 2023; Gao et al., 2022; Jia et al., 2023; Li et al., 2022; Meng et al., 2021; Zhang et al., 2023).

**Oriented Object Detection.** Traditional object detection localizes objects via horizontal bounding boxes, that is, boxes aligned with the $(x, y)$ axes. However, in applications such as aerial imagery (Xia et al., 2018; Sun et al., 2022; Yang & Yan, 2022; Yang et al., 2023a; Yu et al., 2024b) or scene text detection (Liao et al., 2018; Liu et al., 2018; Ma et al., 2018; Wang et al., 2020), extracting the orientation of objects is preferred (Zhou et al., 2022). Thus, oriented object detection predicts an oriented bounding box $(x, y, w, h, \theta)$ instead of the horizontal bounding box $(x, y, w, h)$, where $(x, y)$ corresponds to the center position, $(w, h)$ is the width and height, and $\theta$ indicates the rotation angle (Xu et al., 2024). To achieve this, several works have adapted popular object detection architectures to handle the additional angle prediction (Xie et al., 2021; Ding et al., 2019; Yang et al., 2021b; Han et al., 2021; Li et al., 2023; Xu et al., 2021; Yi et al., 2021; Pu et al., 2023). More recently, Yang et al. (2023a) have developed an approach that learns to detect oriented objects via horizontal bounding box supervision, thus considerably decreasing the annotation cost. Their method, coined H2RBox, leverages rotational covariance to find the minimum circumscribed rectangle inside the HBB. An improved version H2RBox-v2 (Yu et al., 2024b) addresses angular periodicity and achieves OBB-supervised SOTA performance. Moreover, some works have proposed to learn oriented object detection via single point supervision, where only a point is provided for each object. However, the performance of these methods lags far behind HBB and OBB-supervised approaches (Yu et al., 2024a; Luo et al., 2024).

**Boundary and Symmetry Problems.** The angle regression task in oriented object detection introduces two well-known challenges:

1) The **boundary problem** (Xiao et al., 2024; Yang & Yan, 2020; Xu et al., 2024) refers the rotation discontinuities that occur due to angular periodicity. Small differences between prediction and ground truth at the angular boundary will cause a sharp loss increase during training, despite both values being mathematically similar.

2) **Symmetry issues** (Xiao et al., 2024; Yu & Da, 2023), caused by rectangular and squared-like objects, can also negatively impact the learning process. In the case of rectangular objects, a rotation of $\frac{\pi}{2}$ or $-\frac{\pi}{2}$ radians yields equivalent bounding boxes. However, training will penalize a network that correctly predicts $-\frac{\pi}{2}$ if the ground truth is $\frac{\pi}{2}$. Squared objects behave similarly with a $\frac{\pi}{2}$ periodicity instead of $\pi$. This incoherence between loss and box alignment is a source of confusion for the network during training.

Three main strategies have been proposed to mitigate these problems:

a) **Smoothing losses**: These works smooth the loss discontinuities at the angular boundary. SCRDet (Yang et al., 2019a) uses smooth L1-loss, while RSDet (Qian et al., 2021) uses a modulated loss. SCRDet++ (Yang et al., 2022) introduced an IoU factor into the loss term to further address boundary issues.

b) **Angle-coder** methods transform object orientations into representations that are robust to angular periodicity issues. Yang & Yan (2020) have re-framed angle prediction as a classification task, representing orientation using a Circular Smooth Label (CSL) to address the error between adjacent angles. Densely Coded Labels (DCL) (Yang et al., 2021a) later improved performance and speed with a new re-weighting loss based on aspect ratio and angle distance, and addressed symmetric ambiguities of square objects. Yu & Da (2023) transformed rotational periodicity of different cycles into the phase of different frequencies. Their approach, Phase-Shifting Coder (PSC), surpassed previous angle representations as a regression task.

c) **Gaussian-Based Methods**: These approaches model OBBs as 2D Gaussian distributions to solve boundary and symmetry challenges. The Gaussian Wasserstein Distance (GWD) (Yang et al., 2021c) compares predicted and ground-truth distributions, offering an approximation to the non-differentiable IoU loss. Yang et al. (2021d) further refined this with the Kullback-Leibler Divergence (KLD) and later introduced KFIoU (Yang et al., 2023b), a fully differentiable method that approximates the Skew Intersection over Union (SkewIoU) and handles non-overlapping cases without parameters. Gaussian-based approaches provide an elegant solution to boundary discontinuities and symmetric ambiguities. Recently GauCho (Murrugarra-Llerena et al., 2025) proposed to directly regress Gaussian parameters. They enforce positive-definiteness, required by Gaussian definition, by regressing the Cholesky decomposition parameters of the covariance matrix $\Sigma$.

## 3 Motivation

Previous works have attempted to circumvent angular discontinuity issues in different manners. Gaussian-based techniques offer a continuous representation of orientation, while angle-coder methods provide a modular and simple solution via an encoding-decoding mechanism. Nevertheless, current approaches have shown several drawbacks. Yu & Da (2023) state that CSL's performance *could be greatly affected by hyper-parameters; GWD and KLD solve both problems elegantly, but their prediction is relatively inaccurate.* They then add: *without tuning hyper-parameters, CSL and KLD produce rather limited effects.* Moreover, Yang et al. (2023b) state that *the introduction of hyper-parameters makes KLD loss and GWD loss less stable than KFIoU loss*, and that *hyper-parameter tuning may vary across datasets and detectors.* Xu et al. (2024) find that *CSL relies on relatively long encoding, while the choice of coding length for PSC is challenging.* Lastly, Xiao et al. (2024) highlight that *PSC struggles to sustain aspect ratio continuity, particularly when dealing with square-shaped OBBs.*

**The Structure Tensor** is a classic concept in image processing and computer vision introduced by Bigun et al. (1991). It consists of a low-level feature represented as a 2D symmetric matrix that captures the magnitude and direction of local image structures, and has been extensively used in corner and edge detection (Harris et al., 1988; Förstner & Gülch, 1987; Lindeberg, 1998; Brox & Weickert, 2006; Scharr, 2004). Built on the first-order image gradients, structure tensors are inherently continuous over the angular space, and its components transition smoothly and periodically under image rotation. The structure tensor $J$ at each pixel $(x, y)$ is defined as

$$J(x,y) = \begin{bmatrix} \sum J_x^2 & \sum J_x J_y \\ \sum J_x J_y & \sum J_y^2 \end{bmatrix}, \tag{1}$$

where the sums are taken on a neighborhood of $(x, y)$, $J_x$ and $J_y$ are the gradients of the image in the $x$ and $y$ directions, respectively. Let $\lambda_1, \lambda_2$ be the eigenvalues of $J$ with $\lambda_1 \geq \lambda_2$, and $v_1, v_2$ corresponding eigenvectors. The eigenvectors of $J$ represent the principal directions of the local gradients, where $v_1$ points in the direction of the main orientation of the local intensity pattern, and $v_2$ is orthogonal to it. Moreover, the eigenvalues represent the level of intensity variation in the principal directions, and their relative difference encapsulates the anisotropy of the structure, such that:

- If $\lambda_1 \gg \lambda_2$, the local structure is highly anisotropic, thus the gradients are much stronger in one direction.

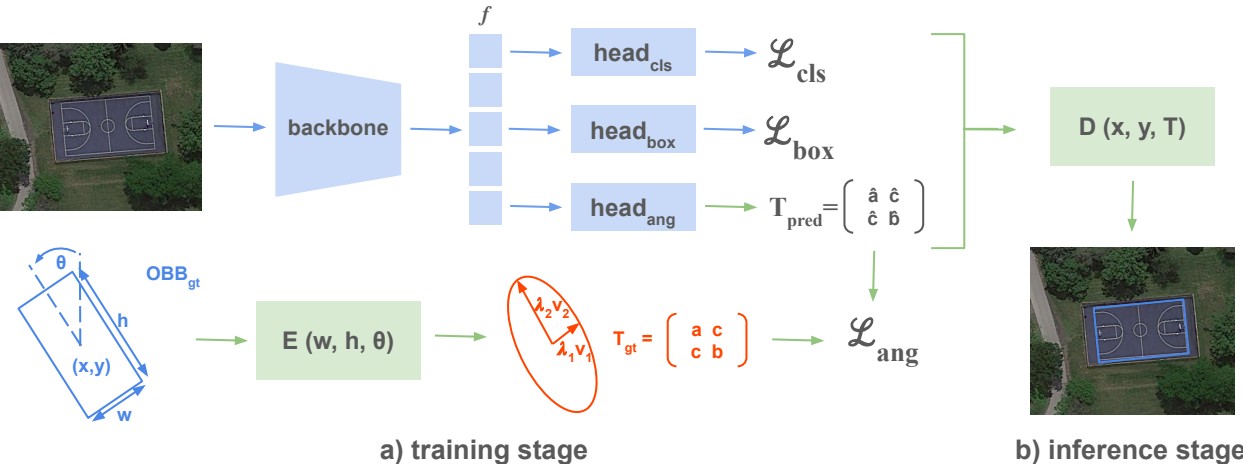

Figure 2: **Relaxed Structure tensor representation in a neural network**. (a) During training, the backbone extracts image features $f$, which are used for classification and regression. The angle head predicts orientation as a structure tensor $T_{pred}$, and the ground truth $OBB_{gt}$ is encoded into $T_{gt}$ for angle loss computation. (b) At inference, $T_{pred}$ is decoded into the standard OBB format $(x, y, w, h, \theta)$. Blue denotes standard detector components, while green highlights our method.

- If $\lambda_1 \approx \lambda_2$, the local structure is isotropic, i.e. the gradients are similar in all directions.

- If $\lambda_1 \approx 0$ and $\lambda_2 \approx 0$ , the structure is homogeneous.

Structure tensors combine the advantages of Gaussian-based and angle-coder approaches. Like Gaussian-based methods, they offer a continuous orientation representation that adapts to different symmetries through the relation between their eigenvalues. Moreover, structure tensors enable a computationally efficient representation that can be implemented in an angle-coder fashion. By predicting orientation similarly to structure tensors, angle regression operates in a space where angular periodicity and boundary discontinuities are naturally avoided. In addition, it requires no hyper-parameters, whereas tuning them is a common challenge in existing solutions, as discussed above. To this end, we introduce the relaxed structure tensor in the following section.

## 4  Method

Let $o_{bb}$ be an oriented bounding box characterized by the usual 5-parameter format $(x, y, w, h, \theta)$, where $(x, y)$ are the center coordinates, $(w, h)$ are the width and height, and $\theta$ is the rotation angle. Moreover, we assume the bounding box format takes the longest side of the bounding box as the width, i.e. $w \geq h$, similarly to previous implementations (Zhou et al., 2022). We introduce the encoding and decoding functions, $E$ and $D$, which map the orientation of the OBB into a structure tensor T and recover the OBB from T, respectively:

$$o_{bb} \xrightarrow{E(w,h,\theta)} T \xrightarrow{D(x,y,T)} o_{bb}. \tag{2}$$

Thus, we propose to directly predict orientation in the structure tensor space to circumvent angular periodicity issues in an angle-coder manner. Figure 2 provides an overview of our approach integrated into a oriented object detector. While the proposed representation adopts the symmetric $2 \times 2$ matrix form of a classical structure tensor, it does not encode the magnitude of image gradients. Instead, we replace gradient magnitudes with the geometric dimensions of the object, i.e. its width and height, alongside its orientation. Although our approach is inspired by structure tensors, we will see that in our context the positive definitiveness can be dropped. We therefore refer to this formulation as a relaxed structure tensor, and to the resulting oriented bounding boxes as Relaxed Structure Tensor Bounding Boxes (RST-BB).

**The relaxed structure tensor.** The proposed relaxed structure tensor is a variation of the traditional structure tensor that enables a continuous representation of orientation in the angular space and reduces ambiguities for redundant cases, such as square-like objects. To this end, we first define the rotation matrix $R_\theta$, denoting a rotation by $\theta$ radians, and the diagonal matrix $\Lambda$, which depends on the width and height of the bounding box:

$$R_\theta = \begin{bmatrix} \cos\theta & -\sin\theta \\ \sin\theta & \cos\theta \end{bmatrix}, \ \Lambda = \begin{bmatrix} w & 0 \\ 0 & \frac{h}{2} \end{bmatrix}. \tag{3}$$

Then, the relaxed structure tensor $T$ is then computed as

$$T = R_\theta \Lambda R_\theta^T = \begin{bmatrix} a & c \\ c & b \end{bmatrix}, \tag{4}$$

where $T$ is parameterized by three values $a$, $b$, and $c$. The eigenvalues $\lambda_1$ and $\lambda_2$ of $T$ represent the height and width of the OBB, respectively, and their eigenvectors $v_1 = [v_{11}, v_{12}]$ and $v_2 = [v_{21}, v_{22}]$ depict the orientation of the object $\theta$.

Furthermore, we highlight two crucial properties of this formulation:

- **Continuity of orientation**: classical structure tensors require positive semi-definiteness, i.e., being symmetric with non-negative eigenvalues. In our case, $T$ is symmetric by construction. Moreover, Weyl's theorem (Weyl, 1946) guarantees the continuity of the eigenvalues of a $2 \times 2$ symmetric matrix, while the Davis–Kahan theorem ensures that the eigenvectors vary continuously with respect to the matrix coefficients (Davis & Kahan, 1970). Hence, we do not directly enforce positive semi-definiteness, but instead leverage these properties to model orientation in a continuous manner.

- **Handling square-objects**: as shown in Equation 3, the height of the bounding box is encoded in $\Lambda$ as half of its value, unlike the width. This avoids the degenerate case where $\lambda_1 \approx \lambda_2$ (i.e., $w \approx h$), which lacks a clear principal direction and would lead the loss to learn an arbitrary orientation. Since $w \geq h$ in our OBB format $(x, y, w, h, \theta)$, the two eigenvalues are guaranteed to maintain a minimum ratio of $\frac{1}{2}$. Consequently, the relaxed structure tensor yields a clear principal orientation and a unique solution even for square-like objects. This reflects reality in most remote sensing contexts; for example, an airplane localized with a square-like OBB still has a unique orientation, corresponding to the direction it is facing. A visual comparison is provided in Figure 1, showing the difference between the traditional OBB format, classical structure tensors, and relaxed structure tensors.

Lastly, given a relaxed structure tensor $T$ and the center coordinates $(x, y)$ of the object, we can decode $T$ back to the original $o_{bb}$. First, we compute the eigenvalues $\lambda_1$ and $\lambda_2$ and eigenvectors $v_1$ and $v_2$ of $T$. As mentioned earlier, the eigenvalues of $T$ characterize the width and height of the bounding box. Furthermore, the orientation of the object is depicted by the direction of the strongest eigenvalue. All in all, we can extract the $w$, $h$ and $\theta$ as

$$w = |\lambda_1|, \ \ h = 2|\lambda_2|, \ \ \theta = \arctan(\tfrac{v_{12}}{v_{11}}), \tag{5}$$

where $v_{11}$ and $v_{12}$ are elements of the eigenvector $v_1$ associated with the larger eigenvalue $\lambda_1$. With known (x,y) values, we can just provide the decoded $o_{bb}$ as $(x, y, w, h, \theta)$.

**Training.** Similarly to Yu & Da (2023), we propose an angle-coder approach where the regression of the angle $\theta$ is learned by the network in the relaxed structure tensor space. In the traditional two-stage detector architecture, a backbone $\mathcal{F}$ extracts high-dimensional features $f$ from an image $I \in \mathbb{R}^{3 \times H \times W}$:

$$\mathcal{F}(I) = f \in \mathbb{R}^{H' \times W' \times D}, \tag{6}$$

where $H$ and $W$ are the image height and width, $H'$ and $W'$ are the height and width of $f$ with the usual decrease in spatial resolution in DNNs, and $D$ is the feature dimensionality. Then, different heads are applied to $f$ for the regression and classification of objects, where typically $head_{cls}$ is tasked with classification, $head_{bbox}$ regresses a horizontal bounding box, and $head_{ang}$ predicts the orientation.

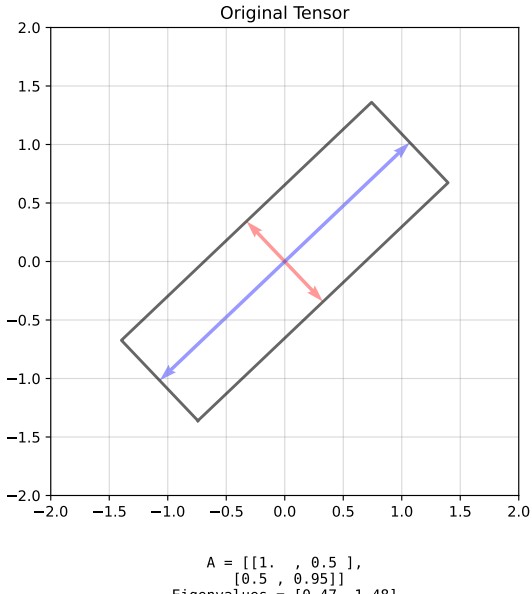
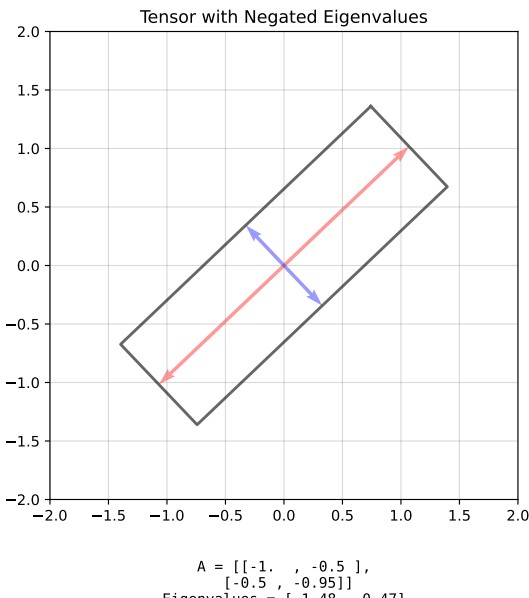

Figure 3: **Comparison between relaxed structure tensors with negative and non-negative eigen-values.** As seen, the two tensors represent the same orientation, despite the negative eigenvalues of the tensor on the right does not hold positive semi-definiteness. Notice the bounding box will keep the longest side as the width, yielding the same box in both cases.

In this context, we integrate the proposed representation into network training by letting $head_{ang}$ predict the three parameters $\hat{a}, \hat{b}, \hat{c}$ that characterize the predicted relaxed structure tensor $\mathbf{T}_{pred}$. During training, the ground truth oriented bounding boxes are encoded into relaxed structure tensor bounding boxes via the encoder $E$. Let $\mathbf{OBB}_{gt}$ represent the ground truth bounding box and $\mathbf{T}_{gt}$ its corresponding relaxed structure tensor, the angle loss $\mathcal{L}_{ang}$ is then computed by the L1-Loss between the predicted and ground truth relaxed structure tensor:

$$\mathcal{L}_{ang} = \frac{1}{N} \sum_{i=1}^{N} \left| \mathbf{T}_{gt}^{i} - \mathbf{T}_{pred}^{i} \right|, \tag{7}$$

where $N$ is the total number of bounding boxes in the training batch. Then, the overall loss is computed as the traditional three-term loss (Xie et al., 2021):

$$\mathcal{L} = w_{cls}\mathcal{L}_{cls} + w_{bbox}\mathcal{L}_{bbox} + w_{ang}\mathcal{L}_{ang}, \tag{8}$$

where $w_{cls}$, $w_{bbox}$, and $w_{ang}$ are the weights for the classification, bounding box regression, and angle regression losses, respectively. While the width and height of the bounding box is encoded in the relaxed structure tensor alongside the orientation, we keep the original $(w, h)$ regression head and only use the tensor as a representation of angular prediction.

**Positive semi-definiteness.** Classical structure tensors are constrained to be positive semi-definite so their eigenvalues are non-negative. In our *relaxed* representation we do not enforce positive semi-definiteness during training: the loss is applied in the encoded space (no eigen-decomposition), allowing the network to learn appropriate value ranges from the ground truth examples. At decoding, we map eigenvalues to the non-negative orthant by taking their absolute values. This produces a valid structure tensor while preserving the orientation carried by the eigenvectors, so explicit positive semi-definiteness constraints are unnecessary. Figure 3 illustrates that tensors differing only by the signs of their eigenvalues retain the same eigenvectors (orientation).

**Differences with Gaussian-based approaches.** Gaussian methods (*i.e..* GWD, KLD, KFIoU) build probability density functions (PDF) from predicted and ground truth $(x, y, h, w, \theta)$, and compare them

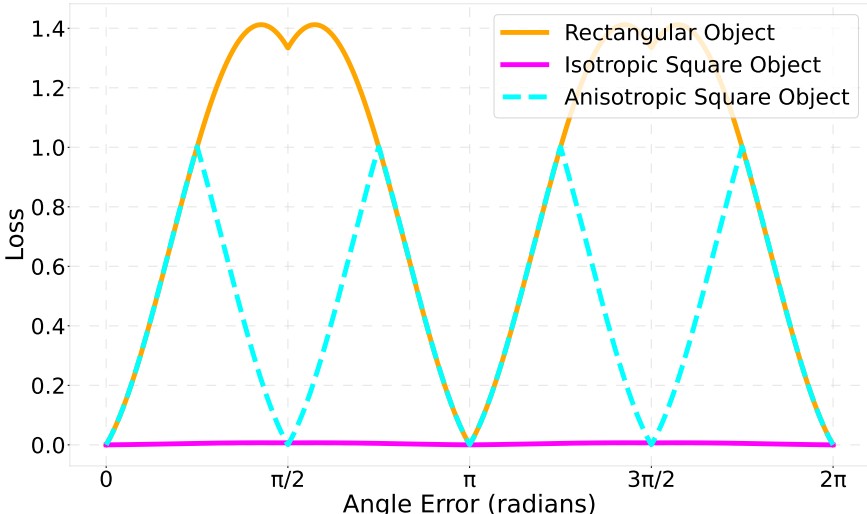

Figure 4: **Loss behavior for square-like objects**. Rectangular objects exhibit $\pi$ periodicity, with loss increasing as angle error grows, which is the desired behavior in order to penalize wrong orientation predictions. However, square-like objects yield consistently low loss values regardless of angle error due to their isotropic nature. Introducing anisotropy by modeling the relaxed structure tensor such that $(\lambda_1 = w, \lambda_2 = \frac{h}{2})$ results in a loss with $\frac{\pi}{2}$ periodicity that achieves high values as prediction error increases.

using probability density distances such as the KL divergence (Yang et al., 2021d) or the Wasserstein distance (Yang et al., 2021c). Hence, during training, they first predict $\theta$ in the angular space and then build the Gaussian distribution. Consequently, their angular prediction occurs in a space suffering from boundary issues. Instead, we treat orientation as a relaxed structure tensor, which does not define a PDF. Thus, our method introduces two key differences:

- Orientation prediction occurs in the relaxed structure tensor space, which is robust to boundary issues.

- We compare relaxed structure tensors directly through L1 Loss, which avoids decoding the RST-BB onto $\theta$ during training. Since we do not use a Gaussian loss, we do not need to enforce the positive definiteness either.

Moreover, recently proposed GauCho (Murrugarra-Llerena et al., 2025), requires $\Sigma$ to be positive definite, as required by Gaussian definition, which is non-trivial for neural networks. Not fulfilling this property will lead to training instabilities and NaN values at the beginning of the training due to the network's random initialization. To address this, they regress the Cholesky decomposition parameters of $\Sigma$ and then compute Gaussian distribution-based losses. In contrast, our method directly regresses parameters $a, b$ and $c$ of a symmetric matrix, and applies L1 loss directly in the proposed representation. This avoids eigendecomposition during training, preventing issues such as `torch.eig()` collapse (NaNs). At inference, only the eigenvectors (which encode the angle) are used; eigenvalues are ignored as $(w, h)$ are predicted by a separate branch, so them being possibly negative is not a source of instability and thus not a concern. Consequently, the novelty of our work lies in its simplicity: no hyperparameters, no eigenvalue decomposition during training, prediction robust to boundary issues and square objects; all while achieving SOTA results. Table 1 illustrates the difference between the proposed approach, Gaucho, and other Gaussian-based methods.

**Pseudo-code.** To highlight the simplicity of our method, we emphasize that with only a few lines of code, one can easily integrate the relaxed structure tensor representation into their own architectures. We illustrate this in Algorithm 1 and Algorithm 2, showing the pseudo-code of the encoding and decoding functions $E$ and $D$, respectively.

| | Gaussian | GauCho | Ours |
|---|---|---|---|
| Predicts orientation in a space that is robust to boundary issues | ✗ | ✓ | ✓ |
| Doesn't build a PDF (thus no positive-definiteness nor additional hyperparameters) | ✗ | ✗ | ✓ |
| No eigenvalue decomposition at training | ✓ | ✗ | ✓ |
| Handles squared objects | ✗ | ✗ | ✓ |

Table 1: Comparison of Gaussian methods, GauCho and Ours.

---

**Algorithm 1:** Encoding Relaxed Structure Tensor Function:

---

**Require:** angle $\theta$, width $w$ and height $h$.
1: Build rotation matrix $R_\theta$ with $\theta$.
2: Build diagonal matrix $\Lambda$ with width and height $(w,h)$.
3: Apply rotation $R_\theta$ to $\Lambda$.
4: **return** parameters $a$, $b$, $c$.

---

## 5 Experiments

### 5.1 Benchmarks and implementation details

We evaluate the proposed approach on different domains. Firstly, we select two extensively used satellite imagery datasets, namely DOTA (Xia et al., 2018) and HRSC2016 (Liu et al., 2017). Then, we evaluate on ICDAR2015 (Karatzas et al., 2015) and MSRA-TD500 (Yao et al., 2012), two scene text detection benchmarks. Moreover, we introduce a novel dataset containing SARS-CoV-2 (COVID-19) tests.

**DOTA** (Xia et al., 2018): DOTAv1.0 is a popular remote sensing benchmark with 2,806 images that contain 188,282 object annotations. Instances are classified into 15 categories, some of which include *plane*, *tennis court* and *small vehicle*.

**HRSC2016** (Liu et al., 2017): The HRSC2016 dataset contains instances of ships in different orientations, both at sea and near land. The training, validation and test sets include 436, 181 and 444 images of different sizes.

**ICDAR2015** (Karatzas et al., 2015): The ICDAR2015 dataset is an oriented scene text detection benchmark that contains 1,000 training images and 500 testing images.

**MSRA-TD500** (Yao et al., 2012): The MSRA-TD500 is an oriented scene text detection dataset with 300 training images and 200 testing images. It contains English and Chinese text annotated at the sentence level.

**C19TD**: We introduce the COVID-19 Test Dataset (C19TD), which contains cellphone-made images of tests to diagnose SARS-CoV-2 (COVID-19). The goal of the dataset is to localize two test landmarks that are essential to determine the outcome and validity of the test: the *well*, i.e. the region where the reactive chemical is applied, and the *result* area, which indicates the outcome of the test. We highlight the significance of this data as it contains noticeably symmetric objects that do not belong to the remote sensing domain. The training, validation and test splits contain 800, 100 and 102 images of COVID tests, respectively. For

---

**Algorithm 2:** Decoding Relaxed Structure Tensor Function:

---

**Require:** relaxed structure tensor $ST$.
1: Compute eigenvalues $\lambda_1$, $\lambda_2$ and eigenvectors $v_1$, $v_2$.
2: Compute the arctan of the elements of $v_1$ to extract $\theta$.
3: **return** angle $\theta$, alongside $w$ and $h$.

---

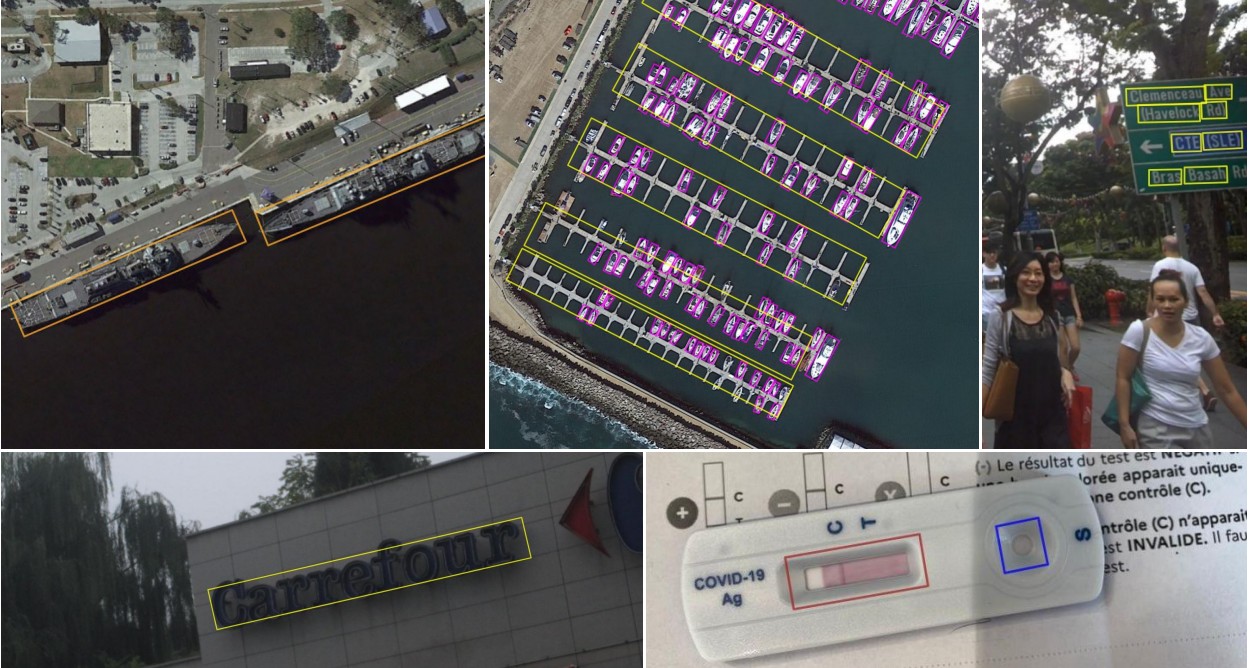

Figure 5: **Qualitative results of the proposed approach on several datasets.** On the top, from left to right, detection examples from HRSC2016, DOTA, and ICDAR2015 are shown. On the bottom, the left image corresponds to the MSRA-TD500 dataset, while the one on the right belongs to the C19TD test dataset. More visualizations and comparisons with other methods can be found in the supplementary.

robustness, we randomly add unrelated natural images without annotations, expanding the train, val and test sets to 1199, 150 and 150, respectively.

**Experimental setup.** We evaluate and compare the structure tensor representation against other SOTA methods. To this aim, we implement our method as an angle-coder object in MMRotate (Zhou et al., 2022), and train several architectures using structure tensors. To make comparisons fair, we compare SOTA methods with the same network architectures and conduct all experiments ourselves via a unified framework. We also implement our method into Gaussian-based models by predicting orientation as a relaxed structure tensor instead of in radians, and then we apply the corresponding Gaussian-based loss. In addition, we train and compare HBB-supervised methods using FCOS with ResNet-50 using different angle representations. HBB-supervised approaches are weakly supervised with horizontal bounding boxes and lack orientation ground truth. All models are trained on a NVIDIA RTX 6000 with a batch size of 2. For the DOTA dataset, we apply the standard pre-processing as per MMRotate (Zhou et al., 2022), generating image crops of 1024x1024 with an overlap of 200 pixels. The remaining datasets are pre-processed by resizing images to 800x512 resolution and augmented with random flips and random rotations. We respect the default hyperparameters of each method in MMRotate and train DOTA over 12 epochs, C19TD over 36 epochs, and HRSC2016, ICDAR2015 and MSRA-TD500 over 72 epochs.

### 5.2 Results

Qualitative detection examples of the proposed relaxed structure tensor representation (RST) on the different evaluation datasets are provided in Figure 5. In this section, we discuss the obtained quantitative results on traditional mAP and the precision of orientation predictions.

**Results on Average Precision.** We compute the standard mean average precision metrics, mAP50 and mAP50:95, on the remote sensing datasets DOTA and HRSC2016 for all methods and report the results in

| | Architecture | Method | DOTAv1.0 | | HRSC2016 | | ICDAR15 | MSRA | C19TD |
|---|---|---|---|---|---|---|---|---|---|
| | | | mAP50 | mAP50:95 | mAP50 | mAP50:95 | mAP50 | mAP50 | mAP50 |
| **OBB supervised** | RetinaNet | Rotated RetinaNet | 69.6 | **40.9** | 83.7 | 52.8 | **65.7** | 64.6 | 95.2 |
| | | GWD | 60.3 | 33.9 | 85.7 | **54.4** | 65.5 | 64.8 | 93.7 |
| | | KFIoU | 69.5 | 37.6 | 84.6 | 48.5 | 65.3 | 64.4 | 95.3 |
| | | **RST KFIoU** | **69.7** | 37.3 | **85.9** | 48.8 | 62.6 | **64.9** | **95.5** |
| | FCOS | Rotated FCOS | 71.1 | **40.6** | 89.0 | 64.0 | 60.4 | 67.9 | 94.9 |
| | | CSL | 71.4 | 40.5 | 89.5 | 58.4 | 55.7 | 68.2 | 90.5 |
| | | KLD | 71.6 | 39.6 | 89.8 | 63.6 | 64.4 | 68.5 | 95.4 |
| | | PSC | 70.2 | 39.2 | 90.0 | 67.6 | 64.2 | 68.7 | 94.2 |
| | | **RST KLD** | **71.7** | 39.3 | 90.1 | 63.3 | 62.5 | 67.6 | 94.5 |
| | | **RST Rotated FCOS** | 71.3 | 39.5 | **90.3** | **67.7** | **66.8** | **69.0** | **95.5** |
| | ViT | ROI Trans | **76.14** | 45.7 | 90.2 | 62.8 | 69.8 | 70.2 | 90.9 |
| | | ROI Trans KFIoU | 75.6 | 42.4 | 90.2 | 62.6 | **70.6** | **70.5** | 95.1 |
| | | **RST ROI Trans** | 75.7 | **45.8** | 90.3 | **68.6** | 65.7 | 70.0 | 95.1 |
| | | **RST ROI Trans KFIoU** | 75.6 | 41.6 | **90.4** | 68.2 | 70.3 | 70.4 | **95.5** |
| **HBB supervised** | FCOS | H2RBox-v2 w/ CSL | 43.9 | 15.5 | 0.80 | 0.37 | 41.4 | 9.1 | 81.4 |
| | | H2RBox-v2 w/ PSC | 71.5 | **39.4** | 89.1 | 57.1 | 41.0 | 24.0 | **95.2** |
| | | **H2RBox-v2 w/ RST** | **72.6** | 39.2 | **89.5** | **59.0** | **46.2** | **32.0** | 95.0 |

Table 2: **Results on the remote sensing datasets DOTAv1.0 and HRSC**, showing mAP50 and mAP50:95 for the proposed structure tensor representation and compared to SOTA methods. Both OBB-supervised and HBB-supervised approaches are reported. For each metric, the best score and the second best score are shown in green and blue, respectively.

Table 2. Additionally, we include mAP50 results for the oriented scene text detection datasets, ICDAR15 and MSRA-TD500, as well as the COVID-19 Test dataset. We compare models across different architectures, including RetinaNet (Lin et al., 2017), FCOS (Li et al., 2023), and ROI Transformer (Ding et al., 2019).

For some methods, such as PSC on DOTA, we were unable to reproduce the reported results from the original article. Despite multiple training runs using the official code and recommended hyper-parameters, fluctuations in performance — also noted in prior works — became evident. To ensure a fair comparison, we maintain consistent experimental settings so that the only differences between our model and state-of-the-art (SOTA) approaches are method-specific parameters. In this context, we emphasize the advantage of our proposed approach, which requires no additional hyper-parameters, simplifying both implementation and reproducibility.

Our method achieved consistent improvements across FCOS models, both OBB- and HBB-supervised. While RetinaNet and ROI Transformer also showed improvements on some datasets, their results exhibited greater variance. Notably, there is a significant drop in performance on scene text detection datasets when transitioning from HBB- to OBB-supervised tasks. This can be attributed to the limited number of images in ICDAR15 and MSRA-TD500. When learning orientation from the redundancy in the data without explicit orientation information, it is expected that these methods are less robust in conditions of limited training dat.

**Precision of Angular Prediction.** While standard mAP provides a general indication of model performance, it does not directly reflect the precision of a network's angular predictions, as better angular prediction might not affect the true positive/false negative count. For example, one might want to address the accuracy of orientation predictions when detecting objects via mAP50, where objects can be reliably detected but may still have imprecise angle estimates. Hence, we introduce two complementary metrics that specifically evaluate orientation accuracy: the mean absolute error ($\text{MAE}_{\boldsymbol{\theta}}$) and the root mean square error ($\text{RMSE}_{\boldsymbol{\theta}}$) of the predicted angle relative to the ground truth for all true positive detections based on mAP50. This way, $\text{MAE}_{\boldsymbol{\theta}}$ provides an intuitive measure of angular error in radians, while $\text{RMSE}_{\boldsymbol{\theta}}$ penalizes large errors more heavily. Note that angular discontinuity and symmetry ambiguities may yield significant angle errors even when predictions are close to the ground truth. To mitigate this, we consider the $\pi$ periodicity of rectangular bounding boxes, calculating the angle error $\delta$ between prediction $\theta_{pred}$ and ground truth $\theta_{gt}$

| Dataset | Arch. | Method | MAE$_\theta$ | RMSE$_\theta$ |
|---|---|---|---|---|
| **DOTA v1.0** | RetinaNet | Rot. RetinaNet | 0.075 | 0.233 |
| | | GWD | 0.086 | 0.223 |
| | | KFIoU | 0.104 | 0.251 |
| | | **RST KFIoU** | 0.101 | 0.230 |
| | FCOS | Rot. FCOS | 0.533 | 0.870 |
| | | CSL | 0.066 | 0.202 |
| | | KLD | 0.547 | 0.877 |
| | | PSC | **0.059** | 0.196 |
| | | **RST KLD** | 0.123 | 0.328 |
| | | **RST FCOS** | 0.068 | **0.188** |
| | ViT | ROI Trans | **0.088** | **0.218** |
| | | ROI Trans KFIoU | 0.112 | 0.242 |
| | | **RST ROI Trans** | 0.089 | 0.223 |
| | | **RST KFIoU** | 0.123 | 0.254 |
| **HRSC 2016** | RetinaNet | Rot. RetinaNet | 0.033 | 0.054 |
| | | GWD | **0.025** | **0.037** |
| | | KFIoU | 0.031 | 0.048 |
| | | **RST KFIoU** | 0.028 | 0.039 |
| | FCOS | Rot. FCOS | 0.023 | 0.040 |
| | | CSL | 0.038 | 0.049 |
| | | KLD | 0.024 | 0.040 |
| | | PSC | 0.019 | 0.028 |
| | | **RST KLD** | **0.018** | **0.026** |
| | | **RST FCOS** | 0.019 | 0.027 |
| | ViT | ROI Trans | 0.032 | 0.043 |
| | | ROI Trans KFIoU | 0.025 | 0.039 |
| | | **RST ROI Trans** | 0.022 | 0.033 |
| | | **RST KFIoU** | **0.018** | **0.028** |

Table 3: **Angular prediction precision.** MAE$_\theta$ and RMSE$_\theta$ scores on DOTAv1.0 and HRSC2016 across different methods. The best score is shown in green and the second-best score in blue.

as follows:

$$\delta = \min \left( \begin{array}{c} \min \left(|\theta_{\text{pred}} - \theta_{\text{gt}}|, |\theta_{\text{pred}} - \theta_{\text{gt}} + \pi|\right), \\ |\theta_{\text{pred}} - \theta_{\text{gt}} - \pi| \end{array} \right). \tag{9}$$

MAE$_\theta$ and RMSE$_\theta$ are then computed using $\delta$ in their standard equations. Table 3 compares the MAE$_\theta$ and RMSE$_\theta$ scores for OBB-supervised methods on the DOTA and HRSC2016 datasets. We exclude circular objects from DOTA, i.e. *baseball-diamond*, *storage-tank*, *roundabout*, as they may have more arbitrary orientations. Results indicate that the structure tensor approach reduces angular error significantly, specially for SOTA methods that are more prone to errors. The average angle precision improvement across evaluated models in DOTA corresponds to 0.176 for MAE$_\theta$ and 0.203 for RMSE$_\theta$, which is equivalent to approximately 10 degrees. More details on this can be found on the supplementary.

**Effect of Anisotropy in the RST Representation.** As shown in Figure 4 and discussed in Section 4, introducing anisotropy to squared objects theoretically reduces the orientation and loss issues caused by the isotropic nature of these instances. Thus, we test this in practice by comparing the results of the relaxed structure tensor representation using isotropic squared objects ($\lambda_1 = \frac{w}{2}, \lambda_2 = \frac{h}{2}$), and anisotropic squared objects with a 2:1 ($\lambda_1 = w, \lambda_2 = \frac{h}{2}$) and 4:1 ($\lambda_1 = 2w, \lambda_2 = \frac{h}{2}$) aspect ratio, respectively. We report the results on DOTA and HRSC in Table 4, in which we can observe how anisotropic relaxed structure tensors considerably improve angle error and mAP50. While a 4:1 anisotropic relaxed structure tensor shows the lowest MAE$_\theta$, the 2:1 tensor achieves the best mAP50 with a minimal MAE$_\theta$ increase that corresponds to 0.34 degrees. This behavior aligns with the idea that the more anisotropic the representation, the easier it

| Dataset | Anisotropy | MAE$_\theta$ | RMSE$_\theta$ | mAP50 |
|---------|-----------|------|-------|-------|
| | Isotropic | 0.083 | 0.200 | 70.4 |
| **DOTAv1.0** | 2:1 | 0.068 | 0.188 | **71.3** |
| | 4:1 | **0.062** | **0.185** | 69.6 |
| | Isotropic | 0.021 | 0.032 | 89.9 |
| **HRSC2016** | 2:1 | 0.019 | **0.027** | **90.3** |
| | 4:1 | **0.018** | **0.027** | 90.1 |

Table 4: **Comparison of structure tensor representations with different levels of anisotropy**. Angle MSE ($MAE_\theta$), angle RMSE ($RMSE_\theta$), and mAP50 are shown for DOTAv1.0 and HRSC2016. The anisotropic structure tensor with a 2:1 aspect ratio yields the best mAP50 scores, with minimal angle error.

| DOTA *plane* class | FCOS | | | ROI Trans | | |
|--------------------|------|-------|-----|-----------|-------|-----|
| | KLD | KFIoU | RST | KLD | KFIoU | RST |
| MAE$_\theta$ | 43.6° | – | 9.7° | – | 16.7° | 11.3° |

Table 5: **Comparison of our approach with other Gaussian-based methods on the square-like object *plane***. Our representation addresses this challenging case and provides a more precise angle prediction, i.e. a lower error with respect to the ground truth.

is for the network to estimate the angle as there is a clear principal direction and a larger penalization of wrong predictions in the loss. This seems to occur until a point is reached, where the extreme anisotropy in the tensor deviates from the real object shape, and mAP scores are affected as a result. We therefore apply $(\lambda_1 = w, \lambda_2 = \frac{h}{2})$ to all objects, regardless of aspect ratio, as it avoids discontinuities and yields the best results in our ablations. The fact that the bounding box format considers the width as the longest side enforces that the principal direction of the relaxed structure tensor is always aligned towards the width, as it corresponds to the greater eigenvalue. Hence, the ambiguity of two possible correct orientations is avoided. This manner, it effectively resolves the square-object issue – unlike GauCho, which states that it *still suffers from decoding ambiguity* in such cases. This is shown in the table below, which reports the MAE$_\theta$ improvement of RST over KLD and FKIoU for the square-like *plane* class in DOTA. Table 5 shows the improvement of our method on the square-object category *plane*, compared to other evaluated Gaussian-based methods.

## 6 Computational complexity

We provide an analysis of the computational complexity of our approach and other methods. To this end, Table 6 reports the FLOPs (in GFLOPs) and the number of parameters for all evaluated methods. As shown, the relaxed structure tensor representation maintains a reasonable complexity for both metrics, comparable to other detectors. Therefore, the proposed approach does not introduce a computational bottleneck compared to the state of the art.

## 7 Qualitative comparisons

We provide additional qualitative comparisons to highlight the impact of the proposed solution. Figure 6 presents detection examples on the DOTA dataset using ROI Transformer, ROI Transformer with KFIoU, and ROI Transformer with the relaxed structure tensor. Additionally, Figure 7 compares results on HRSC2016. As shown, incorporating the relaxed structure tensor into ROI Transformer (with or without KFIoU loss) has a noticeable effect. In HRSC2016, the elongated shape of objects (ships) leads to a scenario where increased angular prediction error prevents detections from becoming false positives, ultimately reducing mAP. The relaxed structure tensor effectively mitigates this issue.

| Method | Architecture | FLOPs | Params (M) |
|---|---|---|---|
| rotated FCOS | FCOS | 206.92 | 31.92 |
| CSL | FCOS | 215.91 | 32.34 |
| KLD | FCOS | 206.01 | 31.92 |
| PSC | FCOS | 207.16 | 31.93 |
| rotated RetinaNet | RetinaNet | 209.58 | 36.13 |
| GWD | RetinaNet | 209.58 | 36.13 |
| KFIoU | RetinaNet | 215.92 | 36.42 |
| **RST (Ours)** | FCOS | 207.01 | 31.93 |

Table 6: **Comparison of models in terms of parameters (in millions) and computational complexity (FLOPs)**. All FLOPs are measured in GFLOPs. ResNet-50 was used as the backbone for all measurements. Our method remains on the lower end in terms of both FLOPs and parameter count compared to other models.

| Method | Original Method | | with RST | | Difference | |
|---|---|---|---|---|---|---|
| | $MAE_{\theta}$ | $RMSE_{\theta}$ | $MAE_{\theta}$ | $RMSE_{\theta}$ | $MAE_{\theta}$ | $RMSE_{\theta}$ |
| KFIoU RetinaNet | 0.104 | 0.251 | 0.101 | 0.23 | 0.003 | 0.021 |
| KLD FCOS | 0.547 | 0.877 | 0.123 | 0.328 | 0.424 | 0.549 |
| FCOS | 0.533 | 0.87 | 0.068 | 0.188 | 0.465 | 0.682 |
| ROI Trans | 0.088 | 0.218 | 0.089 | 0.223 | -0.001 | -0.005 |
| KFIoU ROI Trans | 0.112 | 0.0242 | 0.123 | 0.254 | -0.011 | -0.2298 |
| **Average** | | | | | **0.176** | **0.203** |

Table 7: **Average of angle precision improvement across evaluated models in DOTA**. We compare the angular precision with $MAE_{\theta}$ and $RMSE_{\theta}$ with and without the relaxed structure tensor. As shown, the average improvement of our approach across all methods is 0.176 for $MAE_{\theta}$ and 0.203 for $RMSE_{\theta}$, which translates to approximately 10 degrees.

## 8   Angular error

Section 5 provides an evaluation of angular precision of current methods and the proposed relaxed structure tensor solution via $MAE_{\theta}$ and $RMSE_{\theta}$. We supplement these results by addressing the improvements provided by our solution to specific SOTA methods. Hence, we compare $MAE_{\theta}$ and $RMSE_{\theta}$ of SOTA approaches with and without the relaxed structure tensor approach. This comparison can is showed in Table 7 and Table 8 for DOTA and HRSC2016 respectively. In the case of DOTA, the average improvement in angular precision with the relaxed structure tensor solution is 0.176 for $MAE_{\theta}$ and 0.203 for $RMSE_{\theta}$. This is equivalent to about 10 degrees, which is a considerable correction given the simplicity of our approach and that it requires no hyper-parameters. On HRSC2016, the average improvement is 0.006 for $MAE_{\theta}$ and 0.0114 for $RMSE_{\theta}$. This is equivalent to approximately 0.5 degrees, which is not surprising given that HRSC2016 contains only one class of highly anisotropic objects with a clear principal direction.

## 9   Conclusion

In this chapter, we present a novel angle representation for oriented object detection that effectively addresses angular discontinuities and symmetric ambiguities by encoding orientation as a relaxed structure tensor. It can be easily integrated into existing solutions with minimal effort. Our method is straightforward, efficient, and does not require hyperparameter tuning, unlike many existing approaches. We evaluated our representation across several datasets, including satellite imagery, scene text detection, and symmetric objects, comparing it against state-of-the-art methods. Our approach achieves SOTA performance in average precision across different benchmarks and reduces angular prediction error, outperforming current solutions in some cases.

ROI Trans Ding et al. (2019)  ROI Trans KFIoU Yang et al. (2023b)  RST ROI Trans (Ours)

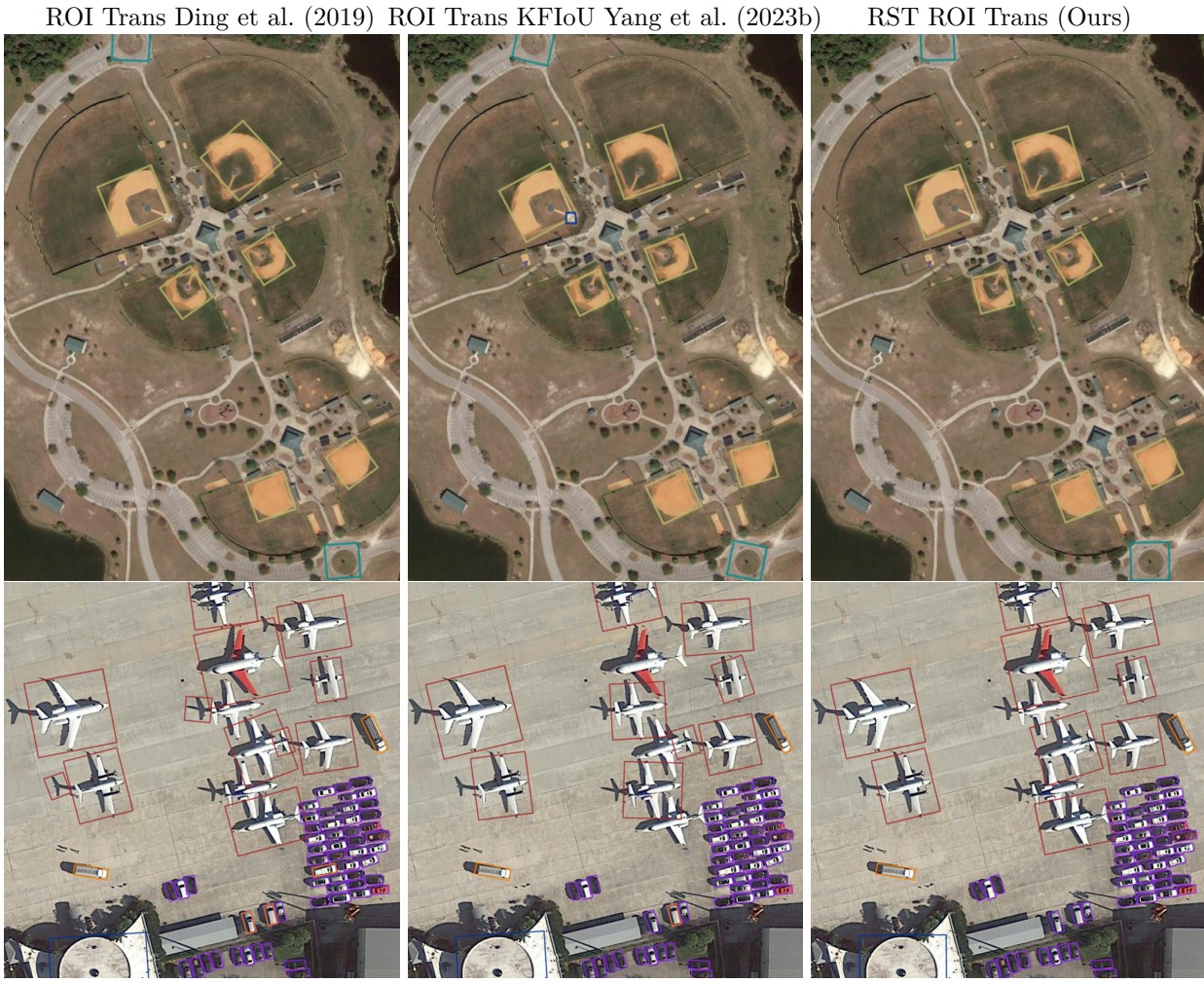

Figure 6: **Qualitative comparison between ROI Trans (Ding et al., 2019), ROI Trans KFIOU (Yang et al., 2023b) and the proposed method (RST ROI Trans), on some DOTA examples.** As shown, the proposed approach predicts a coherent orientation with respect to the detected objects, while the other methods include more error in their orientation predictions.

| Method | Original Method | | with RST | | Difference | |
|---|---|---|---|---|---|---|
| | $\text{MAE}_\theta$ | $\text{RMSE}_\theta$ | $\text{MAE}_\theta$ | $\text{RMSE}_\theta$ | $\text{MAE}_\theta$ | $\text{RMSE}_\theta$ |
| KFIoU RetinaNet | 0.031 | 0.048 | 0.028 | 0.039 | 0.003 | 0.009 |
| KLD FCOS | 0.024 | 0.04 | 0.018 | 0.026 | 0.006 | 0.014 |
| FCOS | 0.023 | 0.04 | 0.019 | 0.027 | 0.004 | 0.013 |
| ROI Trans | 0.032 | 0.043 | 0.022 | 0.033 | 0.010 | 0.01 |
| KFIoU ROI Trans | 0.025 | 0.039 | 0.018 | 0.028 | 0.007 | 0.011 |
| **Average** | | | | | **0.006** | **0.0114** |

Table 8: **Average of angle precision improvement across evaluated models in HRSC2016.** We compare the angular precision with $\text{MAE}_\theta$ and $\text{RMSE}_\theta$ with and without the relaxed structure tensor. As shown, the average improvement of our approach across all methods is 0.006 for $\text{MAE}_\theta$ and 0.0114 for $\text{RMSE}_\theta$, which translates to approximately 0.5 degrees.

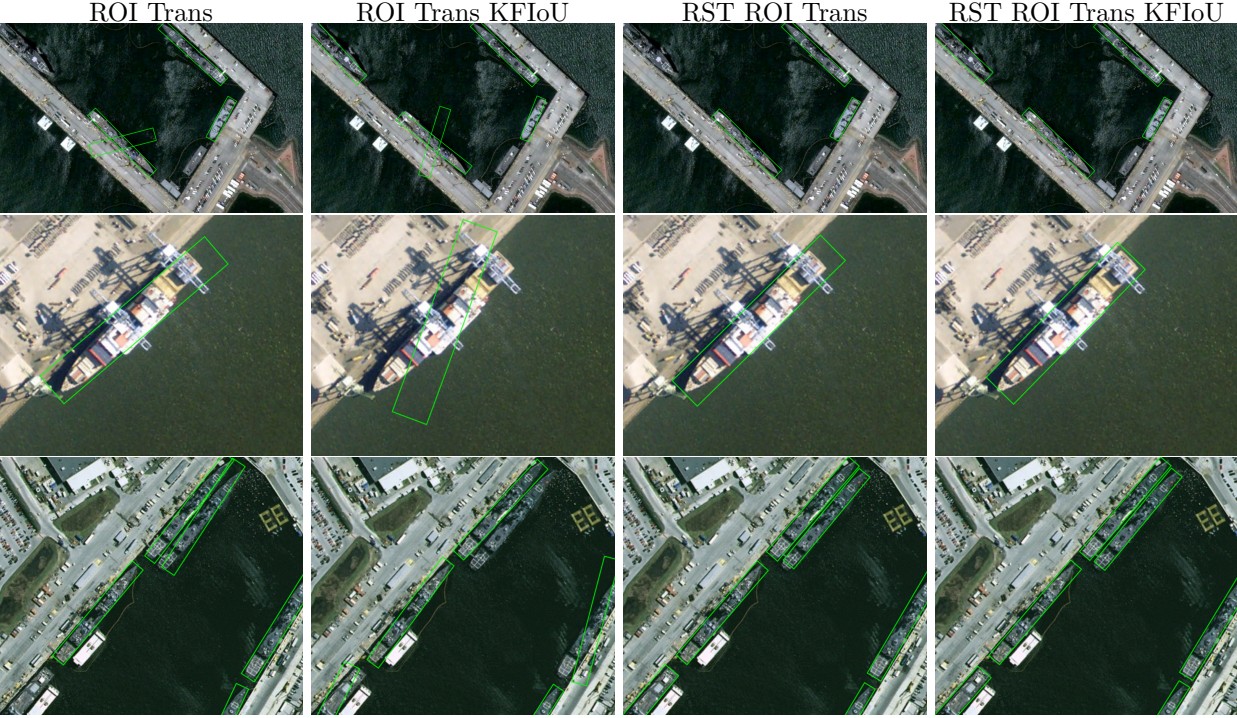

Figure 7: **Qualitative comparison between ROI Trans (Ding et al., 2019), ROI Trans KFIOU (Yang et al., 2023b) and the proposed method (RST ROI Trans and RST ROI Trans KFIoU), on HRSC2016 examples.** The relaxed structure tensor models achieve considerable improvements with respect to the original methods on angle prediction.

ROI Trans        ROI Trans KFIoU        RST ROI Trans

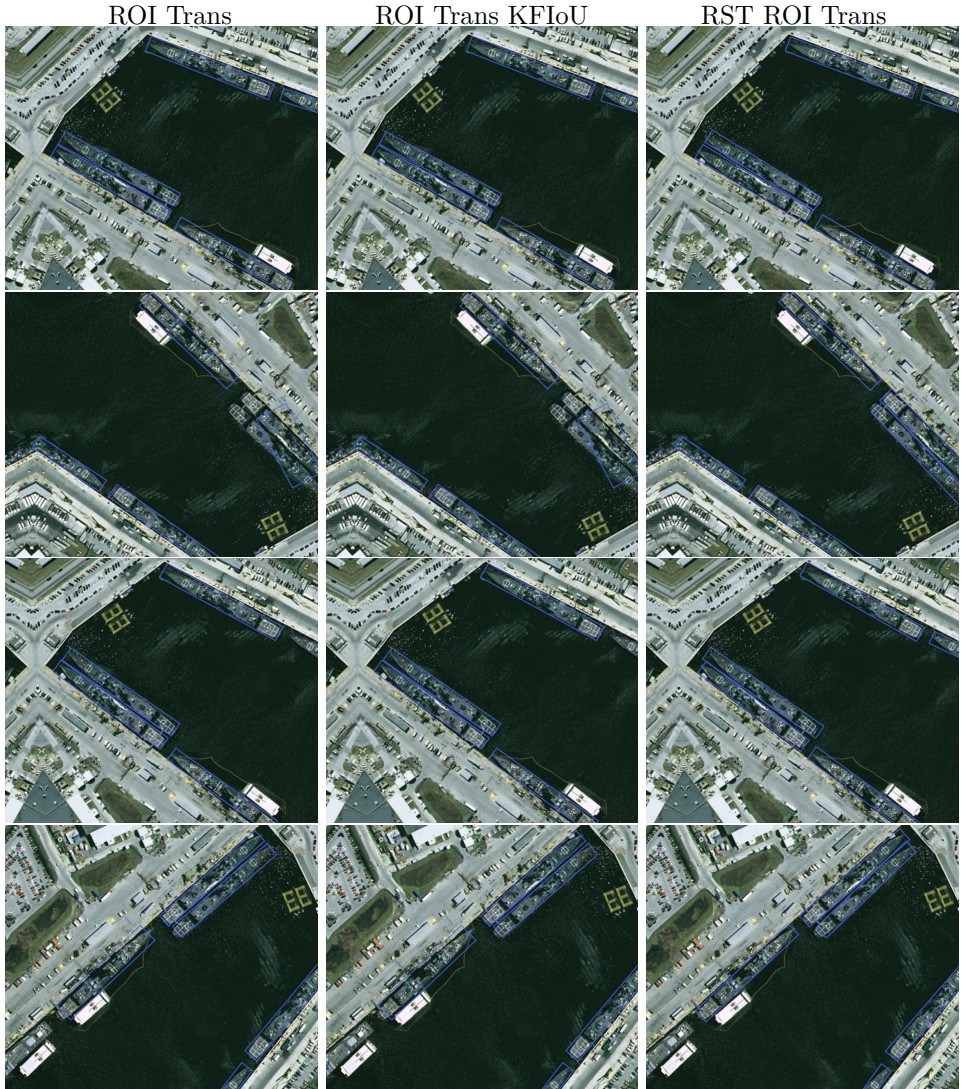

Figure 8: Qualitative comparison between ROI Trans (Ding et al., 2019), ROI Trans KFIOU (Yang et al., 2023b) and the proposed method on HRSC2016, for different rotations of the same image.

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
