# OpenReview forum: "Relaxed Structure Tensor Representation for Robust Ori- ented Object Detection"
_TMLR — Rejected by TMLR_

### Review · Reviewer_bDKv · 2025-10-21

**Summary Of Contributions:**

This paper focuses on the task of oriented object detection, addressing two key challenges: boundary discontinuity issues and symmetry ambiguities. The authors propose an angle-coder approach that demonstrates robustness against angular-related problems. Notably, their method operates without requiring additional hyperparameters.

**Audience:**

Yes

**Audience Explanation:**

Oriented object detection is a fundamental yet challenging computer vision task that provides critical information support for numerous downstream applications.

**Claims And Evidence:**

Yes

**Claims Explanation:**

Experimental results across five datasets confirm the effectiveness of the proposed approach.

**Requested Changes:**

The experimental outcomes appear suboptimal for some small targets, as exemplified by the character recognition case in the upper-right corner of Figure 5.

Regarding efficiency analysis, we recommend the authors compare their method with recent works published in 2024 to date.

---

> ### Author Response · Authors · 2026-01-09
>
> We thank Reviewer bDKv for the insightful comments and suggestions. We agree that detecting small targets is particularly challenging and is an active area of research within the community. We believe that addressing this scenario in detail would be valuable future work.

---

### Review · Reviewer_5npv · 2025-11-25

**Summary Of Contributions:**

This work introduces Relaxed Structure Tensor Bounding Boxes (RST-BB), a new orientation representation for oriented object detection. Rather than regressing angles directly or modeling 2D Gaussian distributions, the method encodes orientation using a 2$\times$2 symmetric matrix constructed from a relaxed structure tensor. The core contributions are:
1. A simple, plug-and-play formulation: angle prediction is replaced with a lightweight 3-parameter tensor head, requiring no additional hyperparameters and integrating seamlessly into existing detectors.
2. Comprehensive empirical validation: the proposed representation is evaluated across five datasets and three detector architectures, demonstrating broad applicability.

**Audience:**

Yes

**Audience Explanation:**

This paper presents a novel formulation for modeling angle prediction, which may provide new insights for advancing oriented object detection.

**Broader Impact Concerns:**

No major ethical concerns.

**Claims And Evidence:**

No

**Claims Explanation:**

The authors claim that their approach “is able to decrease the orientation prediction error, while maintaining SOTA performance.” However, Table 2 shows that this claim does not consistently hold, especially on ICDAR15 and MSRA. When integrating RST into existing detectors, performance frequently drops. For instance, applying RST to FCOS+KLD reduces mAP from 64.4 to 62.5, and incorporating RST into ViT+ROI Trans decreases performance from 69.8 to 65.7. These results suggest that the proposed method may not universally maintain SOTA accuracy across all datasets and architectures, and the claim should therefore be moderated or clarified.

**Requested Changes:**

1. Could the authors provide a more comprehensive analysis of why the proposed method underperforms on ICDAR15 and MSRA? Understanding the failure modes is equally valuable, as it can offer insights into the limitations of the approach and help guide future research.
2. Could the authors provide angular prediction precision results for ICDAR15, MSRA, and C19TD? Since the proposed method appears to degrade overall performance on these datasets, it is important to assess whether RST still improves angular prediction accuracy in these cases.
3. Since the relaxed structure tensor encodes both orientation and the bounding box dimensions, why do the authors retain the original $(w, h)$ regression head instead of directly decoding $(w,h)$ from the tensor itself?

---

> ### Author Response · Authors · 2026-01-09
>
> **The authors claim that their approach “is able to decrease the orientation prediction error, while maintaining SOTA performance.” However, Table 2 shows that this claim does not consistently hold, especially on ICDAR15 and MSRA. When integrating RST into existing detectors, performance frequently drops. For instance, applying RST to FCOS+KLD reduces mAP from 64.4 to 62.5, and incorporating RST into ViT+ROI Trans decreases performance from 69.8 to 65.7. These results suggest that the proposed method may not universally maintain SOTA accuracy across all datasets and architectures, and the claim should therefore be moderated or clarified.**
>
> ***Answer***: We accept the reviewer’s demand and will therefore moderate the claim that our approach improves or achieves SOTA accuracy for all datasets and architectures.
>
> **Could the authors provide a more comprehensive analysis of why the proposed method underperforms on ICDAR15 and MSRA? Understanding the failure modes is equally valuable, as it can offer insights into the limitations of the approach and help guide future research.**
>
> ***Answer***: The proposed method underperforms on ICDAR2015 and MSRA only when combined with ViT-based architectures, and the performance gap remains small (0.3 and 0.1 points below the best-performing methods, respectively). Both ICDAR and MSRA are text detection benchmarks with relatively limited training data. In this low-data regime, ViT architectures, without CNNs’ inductive biases and larger number of parameters, are known to be more sensitive to data scarcity and exhibit higher variance.
> We hypothesize that, under these conditions, integrating our method amplifies this variance rather than providing consistent gains, leading to slightly degraded performance. We propose to clarify this in the main text.
>
> **Could the authors provide angular prediction precision results for ICDAR15, MSRA, and C19TD? Since the proposed method appears to degrade overall performance on these datasets, it is important to assess whether RST still improves angular prediction accuracy in these cases.**
>
> ***Answer***: We can indeed provide the prediction results for ICDAR, MSRA and C19TD. However, we observed that due to the dataset size and complexity, there is barely any difference between angular prediction of different approaches. For example, see the MSRA results shown in the table below. The difference between best performing and worst performing results always corresponds to less than 0.5 degrees. We will also provide the results for MSRA, ICDAR and C19TD, mentioning this observation.
>
> **Table 1: Orientation error on the MSRA dataset.**
>
> | Method                          | MAE ↓ | RMSE ↓ |
> |---------------------------------|-------|--------|
> | Rotated RetinaNet               | 0.117 | 0.152  |
> | Rotated RetinaNet + GWD         | 0.122 | 0.162  |
> | Rotated RetinaNet + KFIoU       | 0.107 | 0.139  |
> | Rotated RetinaNet + KFIoU + RST | 0.113 | 0.148  |
> |                                 |       |        |
> | FCOS                            | 0.117 | 0.153  |
> | FCOS + CSL                      | 0.115 | 0.150  |
> | FCOS + PSC                      | 0.117 | 0.153  |
> | FCOS + KLD                      | 0.118 | 0.154  |
> | FCOS + RST                      | 0.117 | 0.153  |
> | FCOS + RST + KLD                | 0.111 | 0.143  |
> |                                 |       |        |
> | RoI Transformer                 | 0.120 | 0.157  |
> | RoI Transformer + KFIoU         | 0.121 | 0.158  |
> | RoI Transformer + RST           | 0.120 | 0.157  |
> | RoI Transformer + KFIoU + RST   | 0.122 | 0.159  |
>
> **Since the relaxed structure tensor encodes both orientation and the bounding box dimensions, why do the authors retain the original  regression head instead of directly decoding  from the tensor itself?**
>
> ***Answer***: We refer to the answer provided to Reviewer CXBX above. As mentioned, different detection architectures encode width and height in different manners instead of directly regressing the absolute (w, h) values. For example, FCOS generates bounding boxes of different scales and aspect ratios centered on predicted (x,y) and predicts deltas that correct each bounding box at the (top, down, right, left) sides asymmetrically to determine (w,h). We observed how using (w,h) decoded from the RST representation, which is encoded in their absolute values, achieved less precise dimension regression. For this reason, and in consistency with other architectures, we keep a separate bounding box prediction head dedicated to regress (w,h). Moreover, the (w,h) used in the relaxed structure tensor representation is used only to introduce the desired properties into the angle representation.

---

### Review · Reviewer_CXBX · 2025-12-29

**Summary Of Contributions:**

The paper proposes a representation for oriented object detection. While conventional detection uses horizontal bounding boxes to represent objects, certain objects, such as those in aerial imagery or text, are highly rotated. In addition to bounding boxes, oriented object detection requires predicting an angular representation. The paper claims that the proposed relaxed structure tensor representation is novel for this task. This representation can reduce angular prediction error by addressing boundary problems and symmetry issues. Improved performance is reported on relevant benchmarks.

**Audience:**

No

**Audience Explanation:**

I feel that oriented object detection is a relatively narrow field and is less likely to attract broad interest from the community. On the one hand, representations such as per-pixel segmentation are generally more accurate than oriented bounding boxes. With the emergence of strong, general-purpose segmentation models like SAM, the practical need for oriented bounding boxes is further reduced. On the other hand, the applications demonstrated in the paper are quite limited, focusing only on text detection and aerial object detection. For text detection in particular, the method should be directly compared with OCR approaches on standard OCR benchmarks. Moreover, state-of-the-art OCR methods are increasingly adopting pixel-based representations for text.

Overall, the techniques developed in the paper are highly incremental and do not provide new insights or a deeper understanding of the underlying problem.

**Broader Impact Concerns:**

There is no broader impact statement in the submission. I do not consider this to be a critical concern.

**Claims And Evidence:**

No

**Claims Explanation:**

Overall, the paper appears quite incremental, mainly adjusting the classical structure tensor into a relaxed structure tensor representation. I have several technical concerns, listed below:
* Since a standard bounding box prediction head is still used, it is unclear why width and height need to be encoded in the structure tensor. An ablation study that encodes only the orientation (θ) and removes the standard bounding box prediction head seems necessary.
* In conventional object detection, bounding box width and height are predicted in a normalized space with respect to the full image width and height. In contrast, the proposed relaxed structure tensor represents raw width and height values, which may make learning more difficult.
* A key claimed novelty of the paper is using w and h/2 to handle square-like objects. This does not appear to be a systematic solution, as the method would struggle to encode objects where w = h/2. Consequently, this design choice may be overfitted to specific datasets.
* In the experiments, the proposed RST is integrated into several baseline methods, but most reported improvements are marginal (e.g., 0.1–0.2%). While the improvement on the HRSC2016 dataset for mAP_{50:95} is substantial (around 6%), this result requires further explanation and deeper analysis.

**Requested Changes:**

Please refer to my previous technical comments and revise the submission accordingly. In addition, I would like to request stronger justification for studying oriented object detection. In particular, the paper should clearly explain why this problem remains significant, especially given the availability of free-form segmentation approaches that can provide more flexible and accurate object representations.

---

> ### Author Response · Authors · 2026-01-09
>
> **Regarding using the original bounding box head**
>
> ***Answer***: We thank the reviewer for the detailed comments. As mentioned, different detection architectures encode width and height in different manners instead of directly regressing the absolute (w, h) values. For example, FCOS generates bounding boxes of different scales and aspect ratios centered on predicted (x,y) and predicts deltas that correct each bounding box at the (top, down, right, left) sides asymmetrically to determine (w,h). We observed how using (w,h) decoded from the RST representation, which is encoded in their absolute values, achieved less accurate dimension regression. For this reason, and in consistency with other architectures, we keep a separate bounding box prediction head dedicated to regress (w,h). Moreover, the (w,h) used in the relaxed structure tensor representation is used only to introduce the desired properties into the angle representation. We do not extract and decode (w,h) from its angle representation, though they are encoded into the relaxed structure tensor. We propose to specify it more clearly in the main text.
>
> **Using w and h/2 to handle square-like objects.**
>
> ***Answer***: We refer to this in the paper and explain that in the OBB notation, the width w always corresponds to the largest side, enforcing w >= h. For this reason, w = h/2 will never occur as this implies w is lower than h (and thus h becomes the actual w and vice versa in the OBB formulation). The fact that w >= h implies that encoding height as h/2 will always yield a tensor with a strong anisotropy and thus principal direction, as the height will always be encoded as, at least, half of the width (this would be the case where w==h). We propose to further stress this point in the main text inside the Method section.
>
> **Regarding marginal improvements**
>
> ***Answer***: We agree with the reviewer that the improvements are marginal in some cases. This behavior is common in oriented object detection and is largely due to the limitations of the mAP metric. Standard mAP relies on IoU thresholds to determine true positives, which are primarily influenced by the accuracy of the object center and box dimensions. In contrast, moderate orientation errors, especially at mAP@50, often have little effect on IoU and therefore do not change the detection outcome. Thus, methods that primarily improve angle regression frequently yield small gains in mAP, despite providing more accurate orientation estimates.
> The magnitude of improvement is also dependent on the dataset difficulty, as some datasets place greater emphasis on precise orientation. HRSC2016, for instance, contains elongated objects (boats) with large aspect ratios, making IoU more sensitive to angular errors. This particular anisotropic case favors our anisotropic representation, thus explaining the substantially larger gains observed for mAP@50:95 on this dataset.
>
> **Stronger justification for studying oriented object detection.**
>
> ***Answer***: Oriented object detection is a relevant problem particularly in earth observation and remote sensing, where objects are densely packed, arbitrarily oriented, and often elongated. In such scenarios, axis-aligned bounding boxes severely overlap and and lead to ambiguity, whereas oriented bounding boxes provide suitable information  for localization, counting and downstream reasoning. Beyond localization, object orientation itself carries semantically meaningful information. For example, in surveillance or remote sensing time-series, orientation changes can indicate object motion or activity patterns. All of this is supported by the recent literature published in top tier conferences, cited in the main text (KLD , KFIoU , Xu et al. , PSC, Point2rbox, H2RBox, etc.). We strongly believe that this highlights the actual interest of the community and the relevance of the field. Furthermore, our work is inspired from these works, providing a single solution with the benefits of angle-coder methods (simple, modular, flexible) and Gaussian based approaches (elegantly address boundary issues).
>
> Regarding free-form segmentation: While it is true that orientation can be extracted its outputs, oriented object detection offers several practical advantages:
> - Segmentation requires per-pixel prediction, leading to significantly higher computational and memory costs than detection, which is less suitable for large-scale or real-time applications.
> - Segmentation masks do not always yield a well-defined principal orientation. For objects with approximate rotational symmetry (e.g., square or near-square instances), multiple orientations may be equally valid, leading to ambiguity. This is a well-known issue in oriented object detection that our method explicitly addresses.
> - Segmentation relies on dense pixel-level annotations, which are costly to obtain, whereas oriented object detection uses lightweight annotations and can even be trained from horizontal boxes (e.g., H2RBox).

---

> > ### Comment · Reviewer_CXBX · 2026-01-12
> > **bounding box regression and RST**
> >
> > I thank the authors for the rebuttal, which has helped clarify several points.
> >
> > I would like to request further clarification on the role of encoding (w,h) in the RST representation, which appears to be a central contribution of the paper but is never explicitly decoded. In particular, the paper should first report detection performance when (w,h) are directly predicted from the RST representation and compare this with a standard detection head. It turns out that the standard detection head is still required to achieve better performance, it becomes unclear why encoding (w,h) in RST is necessary in the first place.
> >
> > Consequently, an additional ablation in which only the angle is encoded in RST would be needed to isolate and justify the benefit of including (w,h). The authors should provide deeper analysis of the effect of encoding (w,h) in RST—specifically, whether it improves the accuracy of (w,h) prediction, enhances angular estimation, or contributes in some other way.

---

> > > ### Author Response · Authors · 2026-02-16
> > >
> > > We thank the reviewer for the response and agree on the need for a deeper analysis of the role of width and height encoding in the relaxed structure tensor (RST). We address the two concerns below.
> > >
> > > **1. Can (w, h) from RST replace the bounding box head?**
> > >
> > > At first glance, one might assume that the encoded (w, h) values in RST could be directly used to predict bounding boxes, thus removing the bbox regression head. However, this is not straightforward in practice.
> > >
> > > For example, in Rotated FCOS, bounding boxes are predicted by regressing four offsets [top, bottom, left, right] relative to a center point estimated by a centerness head. This formulation allows the model to produce asymmetric boxes around the center, which is important for precise localization.
> > >
> > > In contrast, RST encodes absolute (w, h) values. These do not provide information about how the box dimensions should be distributed across the four directions. A symmetric assumption (i.e., width/2 and height/2 on each side) can be enforced, but this leads to significantly degraded mAP performance, particularly for elongated objects and at higher IoU thresholds. Therefore, the bbox head remains necessary for accurate localization.
> > >
> > > **2. Why encode (w, h) if the bbox head is still required?**
> > >
> > > Although the bbox head is still needed, explicitly encoding (w, h) in RST provides valuable information about object size and aspect ratio. This structural information improves orientation regression.
> > >
> > > To validate this, we conducted an ablation study where RST was encoded with constant anisotropy, using fixed eigenvalue ratios (1:2 and 1:10), instead of the true (w, h) values. In other words, we encode an RST representation that only encodes the angle and the anisotropy and aspect ratio remain constant. The results are shown below for HRSC, ICDAR and CT19 using the rotated FCOS architecture:
> > >
> > > | Method                     | HRSC | ICDAR | CT19 |
> > > | -------------------------- | ---- | ----- | ---- |
> > > | Encoding (w, h)            | 90.3 | 66.8  | 95.5 |
> > > | Constant anisotropy (1:2)  | 78.9 | 66.5  | 88.8 |
> > > | Constant anisotropy (1:10) | 75.3 | 67.6  | 90.0 |
> > >
> > > Including the true (w, h) information shows improved performance, particularly on HRSC and CT19. It is not the case for ICDAR, though the performance is not as degraded. This demonstrates that size and aspect ratio cues contribute positively to orientation prediction and overall detection accuracy.

---

### Decision · Action_Editor_UAPf · 2026-02-06

**Recommendation:** Reject

**Audience:**

Yes

**Audience Explanation:**

The paper addresses a well-motivated problem, namely challenges in oriented object detection due to angular periodicity.

**Claims And Evidence:**

No

**Claims Explanation:**

The majority of reviewers feel that the claims of this paper were not adequately supported. In particular, the encoding of (w, h) alongside \theta into a structure tensor was not adequately justified given that (w, h) are never decoded during training and are predicted by a separate head. The significance of encoding (w, h) in RST remain unclear, and this major conceptual issue was not satisfactorily resolved during the discussion.